# Description and evaluation of REFIST v1.0: a regional greenhouse gas flux

inversion system in Canada

Elton Chan<sup>1</sup>, Douglas Chan<sup>1</sup>, Misa Ishizawa<sup>2</sup>, Felix Vogel<sup>3</sup>, Jerome Brioude<sup>4</sup>, Andy Delcloo<sup>5</sup>, Yuehua Wu<sup>6</sup>, Baisuo Jin<sup>7</sup>

<sup>1</sup>Climate Research Division, Environment and Climate Change Canada, Toronto, Ontario, Canada Correspondence to: Elton Chan (<u>elton.chan@canada.ca</u>)

<sup>2</sup>Center for Global Environmental Research, National Institute for Environmental Studies, Tsukuba, Japan

<sup>3</sup>Laboratoire des Sciences du Climat et de l'Environnement, Chaire BridGES, Gif-sur-Yvette, France

<sup>4</sup>Laboratoire de l'Atmosphere et des Cyclones, UMR8105, CNRS-Meteo France-Universite La Reunion, La Reunion,

France

<sup>5</sup>Royal Meteorological Institute of Belgium, Uccle, Belgium

<sup>6</sup>Mathematics and Statistics, York University, Toronto, Ontario, Canada

<sup>7</sup>Statistics and Finance, University of Science and Technology of China, Hefei, Anhui, China

## 1 Abstract

| 2  | A regional greenhouse gas flux inversion system (REFIST v1.0) is described. This paper                                    |
|----|---------------------------------------------------------------------------------------------------------------------------|
| 3  | provides a comprehensive evaluation of REFIST for three provinces in Canada that include                                  |
| 4  | Alberta (AB), Saskatchewan (SK) and Ontario (ON). Using year 2009 fossil fuel CO <sub>2</sub>                             |
| 5  | CarbonTracker model results as the target, the synthetic data experiment analyses examined the                            |
| 6  | impacts of the errors from the Bayesian optimisation method, inversion time span, prior flux                              |
| 7  | distribution, region definition and the atmospheric transport model, as well as their interactions.                       |
| 8  | The posterior fluxes were estimated by two different optimisation methods, the Markov chain                               |
| 9  | Monte Carlo (MCMC) simulation and cost function minimization (CFM) methods. Increasing the                                |
| 10 | number of sub-regions (unknowns) beyond "optimality" can produce unstable and unrealistic                                 |
| 11 | fluxes for some sub-regions, and does not yield significantly different flux estimates overall. The                       |
| 12 | two optimisation methods can provide comparable, stable and realistic flux results when the                               |
| 13 | transport model error is small (prior $R^2 \sim 0.8$ with synthetic observations), but both methods                       |
| 14 | present difficulty when the transport model error is large (prior $R^2 \sim 0.3$ ). Stable and realistic sub-             |
| 15 | regional and monthly flux estimates for the western region of AB+SK can be obtained, but not                              |
| 16 | for the eastern region of ON without excluding a poorly simulated station. This indicates a real                          |
| 17 | observation-based inversion will likely work for the western region for tracers with similar                              |
| 18 | temporal and spatial emission characteristics to fossil fuel CO <sub>2</sub> [e.g. wintertime CH <sub>4</sub> in Canada]. |
| 19 | However, improvements are needed with the current inversion setup before a real inversion is                              |
| 20 | performed for the eastern region.                                                                                         |
| 21 |                                                                                                                           |

### 22 1. Introduction

| 23 | Continental continuous measurements are useful for understanding and quantifying the             |
|----|--------------------------------------------------------------------------------------------------|
| 24 | regional carbon budgets for the development of emission control strategies to mitigate the       |
| 25 | impacts of global warming. Environment and Climate Change Canada (ECCC)'s Greenhouse             |
| 26 | Gas (GHG) Measurement Program currently operates a network of about 20 ground-based              |
| 27 | stations to accurately measure atmospheric mole fractions of greenhouse gases in Canada. These   |
| 28 | atmospheric mole fractions are the results of the GHG emissions (sources and sinks) coupled      |
| 29 | with the atmospheric transport and chemistry. The goal of this work is to develop an inverse     |
| 30 | modelling approach using these GHG measurements to estimate sources and sinks of GHG in the      |
| 31 | context of national inventories of anthropogenic emissions for the verification of the bottom-up |
| 32 | inventories for specific regions in Canada.                                                      |

33

Bayesian inversion approach for atmospheric applications that incorporates prior fluxes 35 and their associated first guess uncertainties was applied to  $CO_2$  in Enting et al. (1993, 1995) and 36 Fan et al. (1998, 1999). Since then a large number of atmospheric GHG inversion studies 37 spanning over the last two decades have estimated GHG sources and sinks globally including 38 CarbonTracker CO<sub>2</sub> (Peters et al., 2007), CarbonTracker CH<sub>4</sub> (Bruhwiler et al., 2014) and 39 TransCom3 (Gurney et al., 2002). Regionally there have been many inverse modelling studies 40 focusing on Europe (e.g. Bergamaschi et al., 2005, 2010; Stohl et al., 2009; Manning et al., 2011; Rigby et al., 2011; Thompson et al., 2011; Tolk et al., 2011; Cressot et al., 2014) and the U.S. 41 42 (e.g. Zhao et al., 2009; Jeong et al., 2012; Brioude et al., 2011, 2012, 2013; Miller et al., 2013; Gerbig et al., 2003; Kort et al., 2008). Large discrepancies were found in the flux estimates and 43 spatial distributions among studies (e.g. Vogel et al., 2012; Miller et al., 2013), reflecting the 44

- differences in the modelling approaches (e.g. different atmospheric transports, optimization
  methods, etc.) and assumptions (e.g. different prior fluxes and uncertainties, domain definitions,
- etc.).

| 49       | Miller et al. (2014) compared a number of Bayesian models optimized by the cost                   |
|----------|---------------------------------------------------------------------------------------------------|
| 50       | function minimization method (CFM) and the Markov chain Monte Carlo (MCMC) method. The            |
| 51       | conclusion was that the MCMC estimation method produced the most realistic estimates and          |
| 52       | confidence intervals with known bounds. They pointed out inverse modelling approaches based       |
| 53       | on Gaussian assumptions could not incorporate such bounds and often produced unrealistic          |
| 54       | results. For example, emission grids or regions may have known physical constraints (e.g. non-    |
| 55       | negative emissions). Similarly, in Brioude et al (2011), an improvement of the cost function      |
| 56       | method was introduced by using an iterative method to find the median of the posterior            |
| 57       | distribution instead of the mean. When positive (net) fluxes were expected, their method was not  |
| 58       | required to impose any non-negativity constraints on the covariance matrices to ensure positive   |
| 59       | flux results.                                                                                     |
| 60       |                                                                                                   |
| 61       | It is important to point out that many studies applied Gaussian noise to the synthetic            |
| 62       | observations to simulate transport model errors in their sensitivity tests (Stohl et al., 2009;   |
| 63       | Gourdji et al., 2010; Thompson et al., 2011; Miller et al., 2014 and Ganesan et al., 2014). Thus, |
| 64       | when the performance of inversion approaches was compared, the impact of the transport model      |
| 65       | error and bias on the inverse estimates was not fully examined.                                   |
| 66       |                                                                                                   |
| 65<br>66 | error and bias on the inverse estimates was not fully examined.                                   |
|          |                                                                                                   |

The sources of uncertainties in any inverse models should be studied systematically with 68 synthetic data experiments with known fluxes before applying to real observations. This is the

| 69 | motivation for this study in which we assess our inverse modelling approach using different                       |
|----|-------------------------------------------------------------------------------------------------------------------|
| 70 | setups and inversion domains. We characterize the sensitivity and limitations of the various                      |
| 71 | components of the inverse model using a series of synthetic observation experiments that allow                    |
| 72 | us to investigate the impacts associated with individual and combined errors.                                     |
| 73 |                                                                                                                   |
| 74 | We evaluated our inversion setup starting with a target flux distribution that is slowly                          |
| 75 | varying and positive definite (source only). A suitable choice of target is CarbonTracker fossil                  |
| 76 | fuel CO <sub>2</sub> which varies on the monthly timescale. Using CarbonTracker fossil fuel CO <sub>2</sub> model |
| 77 | results with monthly fluxes as the target synthetic observations, we report here on the inversion                 |
| 78 | estimation errors introduced by the prior flux errors, atmospheric transport model errors,                        |
| 79 | optimisation schemes, the sensitivity to the number of source regions optimised, as well as                       |
| 80 | combinations of these sources of errors. This study can provide insights for regional flux                        |
| 81 | estimations for tracers that have similar temporal and spatial emission characteristics to fossil fuel            |
| 82 | $CO_2$ [e.g. wintertime $CH_4$ in Canada with mainly anthropogenic sources (fossil fuel, agriculture              |
| 83 | and waste or landfill) and essentially no wetland emissions]. Other tracers such as $N_2O$ and $SF_6$             |
| 84 | which are predominately contributed from the anthropogenic sources with small seasonality can                     |
| 85 | potentially be used for flux inversion following the methodology developed in this study.                         |
| 86 |                                                                                                                   |
| 87 | The term "posterior error" will be used wherever appropriate throughout the text to                               |
| 88 | represent the estimation error [relative percentage difference of the posterior flux and the target               |
| 89 | flux, i.e. (posterior flux – target flux)/target flux) x 100%]. The contributions and the interaction             |
| 90 | of the different error components including the errors of the inversion procedure, prior flux and                 |
| 91 | transport model are examined using sensitivity experiments. However, in the real observations-                    |
|    |                                                                                                                   |

based inversion, the magnitude and sign of the errors are often not known and often treated as

- part of the total estimation uncertainty. This study will show that uncertainty of the flux estimates
- could often be unrealistically small. The sensitivity of the estimation error (when the truth is
- 85 known in synthetic experiments) and uncertainty (when the truth is not known in reality) needs to
- be closely examined in any inversion setup.

#### 98 2. Methods

| 99  | In this study, the components of atmospheric inversion include 1) the synthetic                   |
|-----|---------------------------------------------------------------------------------------------------|
| 100 | observations (target), 2) a Lagrangian particle dispersion model (LPDM) run in backward           |
| 101 | (adjoint) mode, 3) assimilated meteorological fields used to drive the LPDM, 4) prior spatial     |
| 102 | distributions of emissions, 5) a method to estimate the baseline (background influence) of the    |
| 103 | observations, and 6) a statistical technique to minimize any differences between prior and target |
| 104 | mole fractions. The observed atmospheric $CO_2$ mole fractions were not used, instead, synthetic  |
| 105 | observations (no land/ocean sink and no biospheric contributions) were simulated from monthly     |
| 106 | fossil fuel CO <sub>2</sub> fluxes that were extracted from the outputs of the global model NOAA  |
| 107 | CarbonTracker release version 2011 (CT2011). Figure 1 shows a schematic of one set (III) of       |
| 108 | inversion experiments. The impacts of the components to the flux estimates as highlighted in      |
| 109 | gray boxes are the focus of this study. The details are described in the following sub-sections.  |
| 110 |                                                                                                   |
|     |                                                                                                   |

## 2.1. Observation stations and inversion domains

Seven existing surface GHG monitoring stations were selected as a test bed for evaluating 113 the inverse modelling approach. These seven GHG stations summarized in Table 1 are located in 114 the three Canadian provinces of Alberta, Saskatchewan and Ontario that together account for

| 115 | close to 70% of Canada's total GHG emissions annually (ECCC, 2015). In 2013, $CO_2$                                   |
|-----|-----------------------------------------------------------------------------------------------------------------------|
| 116 | contributed 78% (and $CH_4$ contributed 15%) of the national total GHG emissions of 726                               |
| 117 | megatonnes (Mt) of CO <sub>2</sub> equivalent (ECCC, 2015). The majority of Canada's national total                   |
| 118 | anthropogenic GHG emissions resulted from the combustion of fossil fuels at about 80% and the                         |
| 119 | remaining portions were contributed from industrial processes, waste incinerations, agricultural                      |
| 120 | activities and landfills.                                                                                             |
| 121 |                                                                                                                       |
| 122 | In this study, the inversion was done separately for the western region of Alberta and                                |
| 123 | Saskatchewan provinces, and the eastern region of Ontario using seven region definitions as                           |
| 124 | shown in Fig. 2a-g to investigate whether there are problems or benefits in estimating the fluxes                     |
| 125 | from a large number of sub-regions.                                                                                   |
| 126 |                                                                                                                       |
| 127 | 2.2. Prior fluxes                                                                                                     |
| 128 | Two sets of fossil fuel CO <sub>2</sub> fluxes (CT2010 and CT2011 for year 2009) were used as prior                   |
| 129 | and target (known "truth") fluxes and summarized in Table 2, which includes the monthly and                           |
| 130 | annual provincial totals. The fluxes were uniformly re-distributed to $0.2^{\circ} \ge 0.2^{\circ}$ from the original |
| 131 | resolution of 1° x 1° to be folded into the emission sensitivity fields from FLEXPART (next                           |

Section). For visualization, the gridded fluxes were aggregated into sub-regions as shown in Fig.

2. Year 2009 country and global totals (by fuel type) were extrapolated from the 2007 Carbon

Dioxide Information Analysis Center (CDIAC, Boden et al. 2013) used for the CT2010 fossil

fuel fluxes (CarbonTracker, 2010). Open-source Data Inventory for Anthropogenic CO<sub>2</sub>

(ODIAC, Oda and Maksyutov, 2011) emissions are spatially distributed using many available

"proxy data" that explain spatial extent of emissions according to emission types (emissions over

- land, gas flaring, aviation and marine bunker). CarbonTracker combined the ODIAC emissions
- with CDIAC emissions to generate CT2011 fossil fuel fluxes (Andres et al., 2011,
- CarbonTracker, 2011).
- **2.3. Transport**

The European Centre for Medium-range Weather Forecasts (ECMWF) operational wind

fields at T799 spectral resolution were used to drive the Lagrangian particle dispersion model

FLEXPART (Stohl et al., 2005). The ECMWF modelled data were retrieved with a temporal

resolution of 3-h (analyses at 0000, 0600, 1200, and 1800 UTC; forecasts at 0300, 0900, 1500,

and 2100 UTC) for two domains. The inner domain has a horizontal resolution of  $0.2^{\circ} \ge 0.2^{\circ}$  on

the Gaussian grid over Canada and the US (180°W to 0°E and 20°N to 90°N). The outer domain

is a global grid with resolution of 1° x 1°. Both grids have 91 vertical levels. The FLEXPART

model was used to simulate the 5-day transport history (retroplume) of the fossil fuel CO<sub>2</sub> mole

fractions at each station location. The model calculated the trajectories of 5,000 particles from the

intake height at each station location daily at 21:00 UTC (14:00 to 16:00 LST depending on time

zones) representing afternoon well-mixed condition near the surface.

FLEXPART retroplume spatial distributions were output as 30-minute averages on a  $0.2^{\circ}$ x  $0.2^{\circ}$  grid. The retroplumes were then summed up for the entire 5 days for each time point (21:00 UTC daily) of particle release. The retroplume is the residence time of the plume per grid cell divided by the air density that has units of s kg<sup>-1</sup> m<sup>3</sup>. The footprint layer of the retroplume for FLEXPART is fixed at the standard 100 m layer adjacent to the Earth's surface (Stohl et al., 2005). The modelled fossil fuel CO<sub>2</sub> mole fractions were constructed by multiplying the

| 161 | retroplume distribution (footprint) with the monthly prior fossil fuel $CO_2$ fluxes at $0.2^\circ \times 0.2^\circ$ in |
|-----|-------------------------------------------------------------------------------------------------------------------------|
| 162 | kg s <sup>-1</sup> and summed up over all grid cells (plus the baseline or the contribution from prior to the 5-        |
| 163 | day simulation period, described below) to yield the time series of modelled fossil fuel CO <sub>2</sub> mole           |
| 164 | fractions at the measurement station (Stohl et al., 2003, 2009; Cooper et al., 2010). The mean                          |
| 165 | footprint of the seven stations for January through December 2009 is shown in Fig. 3 to reveal                          |
| 166 | areas where the surface emissions can likely be constrained using the selected stations.                                |
| 167 |                                                                                                                         |
| 168 | 2.4. Baseline estimations                                                                                               |
| 169 | The station-specific baseline in this context represents the influence from emissions 5                                 |
| 170 | days earlier and beyond. The mole fractions of the fossil fuel $CO_2$ were sampled from the                             |
| 171 | CT2011 predicted global fossil fuel CO <sub>2</sub> field at the positions (latitude, longitude and altitude) of        |
| 172 | 5000 particles at the end of the 5 <sup>th</sup> day backward simulation for each station released at 21:00             |
| 173 | UTC daily to obtain 5000 mole fraction values. These 5000 mole fractions were averaged to                               |
| 174 | represent the mean baseline for each release time point. The station-specific baseline time series                      |
| 175 | was subsequently subtracted from the synthetic observations that were sampled from CT2011 for                           |
| 176 | each station. This allowed us to infer fluxes over the region of interest. Errors in the baseline                       |
| 177 | estimation were treated as a part of the transport error when CT2011 mole fractions were used as                        |
| 178 | the "target".                                                                                                           |
| 179 |                                                                                                                         |
| 180 | 2.5. Two Bayesian inversion methods                                                                                     |
|     |                                                                                                                         |

- In addition to the more common analytical-based CFM approach, we include a
- simulation-based method for flux estimations, MCMC. Sensitivity analyses of the two inversion

| 183 | methods in terms of percentage differences between the posterior estimates and the target fossil      |
|-----|-------------------------------------------------------------------------------------------------------|
| 184 | fuel $CO_2$ fluxes are assessed. It is not the intention to compare which one of these two methods is |
| 185 | more superior to the other, but to evaluate the sensitivity of the results using different inversion  |
| 186 | methodologies and assumptions.                                                                        |
| 187 |                                                                                                       |
| 188 | Note that matrices and vectors are in <b>bold</b> and italic throughout this paper, whereas scalar    |
| 189 | quantities are in italic font. Inversion was done separately for the western and eastern domains,     |
| 190 | and separately for every three months of 2009 that is January-March, April-June, July-September       |
| 191 | and October-December.                                                                                 |
| 192 |                                                                                                       |

The prior gridded fluxes of fossil fuel CO<sub>2</sub>,  $\{x_{g,p,t}\}$  were re-distributed from the original 194 1° x 1° uniformly to the same spatial resolution of 0.2° x 0.2° as the emission source sensitivities  $\{M_{g,p,t,s}\}$  (or footprints), where the subscripts are, g for a given grid cell in sub-region p, station s 195 196 and time t.  $x_{a,p,t}$  is the gridded emission field over sub-region p at time t. The footprints vary in 197 space, time and stations. The modelled mole fractions in our experiments were limited to 21:00 198 UTC daily (14:00 to 16:00 LST depending on time zones) in January through December for 2009 199 to avoid temporal correlation and night time processes. Two regions of interest are the two 200 neighboring provinces of Alberta and Saskatchewan (western region), and separately, the 201 province of Ontario (eastern region) in Canada. Any remaining contributions from outside of the 202 inversion region but within the 5-day integration period were subtracted from the synthetic 203 observations for each station in addition to the station-specific baseline time series.