# Peer review of "Description and evaluation of REFIST v1.0: a regional greenhouse gas flux"

_Geoscientific Model Development, 2016_

## Referee Comment (RC1) · Anonymous Referee #1 · 3 Jan 2017

This study investigates the performance of a regional inversion of anthropogenic CO2 emissions using synthetic experiments varying the prior fluxes, transport model and optimization method, and inversion setup. Conclusions are drawn regarding the optimal number of regions to be optimized, and the relative importance of transport model uncertainties. However, as will be explained further below, it is not clear what we learn in the end. Most of the findings that are presented will depend on the specific setup of the inversion that is chosen. However, since this setup is far from realistic – the practical significance of the results for regional inverse modeling of CO2 using real data remains unclear. In my opinion, this point will have to be addressed clearly in the revised version of the manuscript to make this manuscript acceptable for publication.

In addition, to improve readability I recommend that the authors focus only on the main findings, which could substantially reduce the size of the paper.

GENERAL COMMENTS

The authors recommend that an inversion system is first tested in a synthetic environment before it is applied to real data. I fully agree to this, however, since the outcome of such an experiment depends on the specific details of the setup it should be realistic in the sense that the same setup could directly be applied to real data. This is clearly not the case for the setup that is presented here, since it only addresses anthropogenic emissions of CO2, ignoring the natural component of the regional carbon cycle. In addition, the boundary conditions of the regional domain that is optimized are assumed to be known exactly. Since the regional biospheric fluxes are expected to be the main uncertain component, it is unclear to me how this inversion is supposed to work when applied to real data. It is mentioned that the methodology could be applicable to wintertime CH4 fluxes. But if this is the application that the authors have in mind then why perform a test inversion for fossil CO2 instead of CH4?

The authors comment on the estimated posterior uncertainties in relation with the actual deviations from the predefined true fluxes, concluding that the estimates are too optimistic. However, this conclusion depends on how the assumed a priori flux and data uncertainties reflect the actual errors. It seems that no effort was made to analyze the statistics of the difference between for example CT2011 and CT2010 before specifying the a priori flux covariance. The same is true for the model-data mismatch. In this case, how can the method of calculating posterior uncertainties be blamed of inconsistencies? I wonder also how representative the difference between CT2011 and CT2010 is for uncertainties in fossil fuel CO2 emissions. They can certainly not be considered independent estimates of fossil fuel fluxes.

Conclusions are drawn regarding the relative performance of Bayesian cost function minimization and the MCMC method. However, how can different methods to find the

solution of an inverse problem be compared if they are applied to different inverse problems? A clearer distinction should be made between optimization method and inversion setup. If the MCMC method was applied to the same optimization problem, one would expect to find the same solution – unless one method fails to find the optimum, e.g. because of non-linearity. In this case, at least the CFM inverse problem seems linear, so it should be capable of finding the right solution. The difference between MCMC and CFM seems more in the assumed statistics (inverse Gamma versus Gaussian). But then if MCMC performs better it is probably because of a different weighting of outliers. Further analysis is needed to gain understanding of what is causing the difference between the two methods.

The discussion about aggregation errors and the optimal number of regions seems to have been influenced by the treatment of a priori flux uncertainties. If a region is split up in two equal parts, then care should be taken to specify the uncertainty of the separate regions such that they don't alter the uncertainty of the combined region. In this study, it seems that 100% uncertainty is assumed regardless of the size of a region. But then if the individual fluxes are assumed to have independent uncertainties, the aggregated uncertainty of the whole region will go down as it is split up into a larger number of sub-regions. This is because the errors of the sub regions will partially cancel out in the regional integral (with the square root of the number of regions). Unless this issue is dealt with carefully, it will confuse any inferences about aggregation errors.

SPECIFIC COMMENTS

line 10: 'Increasing the number ...' Does this mean that none of the set ups is significantly different from 'unstable' and 'unrealistic'?

line 13: 'prior R2 ∼0.8' Wouldn't it be better to quantify transport model error using the true fluxes (otherwise it is unclear which part of R2 is due to the prior flux uncertainty).

line 16: 'a poorly simulated station' Why not just mention the station here?

line 19: 'improvements are needed with the current inversion setup ...' It seems that the data availability is the problem. This sentence suggests that an improved setup can compensate for missing measurements. It may be that the problem is in the word 'setup', but then this should be formulated more clearly.

line 28: 'These atmospheric mole fractions fractions ...' Here reality is described as if it is a model. Please change the formulation to avoid confusion.

line 75: What is a positive definite flux distribution. The term 'Positive definite' refers to a symmetric matrix.

line 81: the spatiotemporal distribution of regional CO2 and CH4 fluxes are rather different.

line 94: The sensitivity of the estimation error and uncertainty to what?

line 128: What makes these two estimates suitable as prior and truth?

line 130: How were the fluxes redistributed to 0.2 x 0.2?

line 157: Something must have gone wrong with this definition of retroplume.

line 161: The unit of the multiplication is kg/kg*m3 i.o. mole fraction

line 162: Prior means before here, rather than a priori, right? Please avoid confusion here.

line 174: How were the station specific baseline time series quantified? It sounds like you calculate the baseline contribution from the back plume initializations which you then subtract. However, in reality you don't have the 'true' initial values.

line 209: but you apply an a priori constraint to lambda, which is effectively equivalent to the regularization in CFM.

line 222: If no regularization term is used in MCMC than how can it use the same prior error?

line 249: Why is it necessary to compare means? Since in the Gaussian assumption mean and median are the same, you might as well compare medians.

line 251: Why do you take the average of intermediate solutions in the iterative optimization? Shouldn't the optimum solution be the end point to which the iterative chain converges?

line 252: Since lambda is defined as time independent this sentence is not needed anymore (better would be to state explicitly that lambda is time independent at the point where it is defined).

line 263 - 265: But those scaling factors are intermediate solutions in a optimization process, therefore they are not independent optimal solutions of the inverse problem. For this reason, I don't see how the statistics of the scaling factors could represent the posterior flux uncertainty.

line 321: I would rather call the prior flux error a disaggragation error, since it is mostly the spatial disaggragation which is different between CT2010 and CT2011.

line 338: It is still no cler to me what causes the baseline error in this inversion.

line 344: 'an example of one inversion experiment', which inversion experiment?

line 363: Why is the same representation error of 30% used for all sites, when some sites are easier to simulate by the model than others? By the way, 30% is 30% of what? The deviation from the baseline?

line 374: 'representS'

line 471: This paragraph refers to the same figure as the one before, but why then do you explain the figure here and not before?

line 483-484: I think it is clear that an improved fit to assimilated data is not the right way to validate inversion-estimated fluxes. What is done, however, is to test whether the optimized model does a better job simulating independent data (i.e. that were

not used in the inversion). You could do this test as well, which would yield a more meaningful answer regarding inversion validation.

line 493-494: It is not clear how the seasonal variation can become larger if the state vector is time independent.

line 517: But in set I the CT2010 fluxes were used.

line 546: Whether or not the results can be considered significantly different obviously depends on the spatiotemporal scale over which fluxes are integrated. The scale should be specified more clearly.

line 547-549: This formulation is too vague and needs to be supported by actual numbers.

line 553-551: It is not clear why going from 1 month to 3 months is increasing the observational constraints. What is expressed on the y-axis is the annual flux error. Whether this is composed of 4 block of 3 monthly fluxes or 12 blocks of monthly fluxes doesn't make a difference regarding the number of data that are used. The only difference would be difference in the temporal degree of freedom of the fluxes, but this is not the way it is explained in the text. This comparison needs to be explained more clearly.

line 559: It is unclear why the transport error statistics would be so different for two regions that are not very different regarding transport

line 561-563: Another possible strategy to do what? Please explain more clearly.

line 572-573: But if the prior flux is the truth, then increasing the observational constraint is expected to increase the posterior flux error (an 'inversion' without any observations will yield the correct flux).

line 610-612: Statistically it is not expected that the true flux is always within the 2 sigma interval. If the actual error exceeds the posterior uncertainty this could simply mean that the prior flux uncertainty doesn't properly reflect the prior flux error, or that

the model-data mismatch doesn't properly account for transport model error. Both of these options are likely, given that ad-hoc assumption on these uncertainties were made in the inversion set-up.

line 622-624: Flux error contributions occasionally cancelling each other out are not a sign of non-linearity. If the inversion is linear, as seems to be the case here, you would actually expect the error contributions to add up. If they don't, it raises the question why this happens.

line 630-631: Unrealistic results for some months and sub regions are expected when increasing the degrees of freedom beyond the point that can be resolved by the data.

line 635: A reference is needed here.

line 646: But systematic differences in simulated concentrations during nighttime are probably not just caused by horizontal resolution.

line 635: If the representation error is not a concern, does this mean that the 30% uncertainty that is assumed was too large?

line 685: Given the difficulty to separate the contribution of aggregation errors from other errors, how do you know that using CH4 as a prior causes the largest aggregation error? Where has this been shown?

line 698: What is meant with 'degree of spatial resolution'?

line 726: What do you mean by optimization procedure error?

line 776: What is the difference between equation 1 and A1? (same question for 2 and A2)

line 847-877: This is not right. The most commonly used inverse modelling methods define the state vector elements as random variables.

line 917-919: This is not right. The size of matrices in analytical inversions is limited

by computer memory, but this happens for state vector sizes that are much larger than 'only a few parameters'.

line 925-927: In many cases the inverse problem is approximately linear, and the statistics not far from normal. In this case, you won't have multi modal distributions and the use of means, or medians together with an estimate of the width of the distribution is perfectly fine.

TECHNICAL CORRECTIONS

line 5: 'analysis' i.o 'analyses'

line 52: remove 'with known bounds'

line 129: 'are summarized'

line 188: The GMD formatting policy is to use bold roman for matrices.

line 315: 'using' i.o. 'used'

---

## Referee Comment (RC2) · Anonymous Referee #2 · 13 Jan 2017

The manuscript describes a regional inverse modeling system that uses atmospheric observations to estimate GHG fluxes. Synthetic experiments are performed to evaluate the system. A number of specific aspects need to be addressed before I can recommend accepting the manuscript for publication.

Main comments:

Regarding the number of regions: I fully agree with referee #1 in that a prior error of 100% for different regions with a changing number of regions will result in a decrease of the uncertainty of the spatially integrated fluxes, as the errors are assumed uncorrelated. One possible method would be to inflate the variance accordingly such that the prior error of the spatially aggregated flux does not change with the number of regions. However this has an impact on the comparisons between flux estimates using a different number of regions to be solved for. This might also be part of the reason that posterior estimates shown in Figure 7 are closer to the prior when the number of regions is larger.

The method to calculate the baseline is a bit problematic: Using CT2011 predicted fossil fuel $CO_2$ extracted at locations 5 days before arrival at the observation site does not separate the outside-domain influence from inside-domain influence, where domain means the regional domain of interest. A better method would be to sample the 3d $CO_2$ field at the locations when trajectories first leave that regional domain. To assess the impact, at least a map showing a distribution of the locations of the trajectories at the time step 5 days prior to the measurement should be provided (may be included in an appendix).

Flux error is referred to in the manuscript as the difference between the posterior flux and the target flux (true flux). However, in inverse modeling, usually the statistical uncertainty in the posterior flux estimate is used as an estimate of the expected error in the retrieved flux. In a synthetic experiment, the actual difference between retrieved and true fluxes can be regarded as a realization of this posterior uncertainty. Note that the flux error as referred to in this manuscript is thus expected to be within the 1-sigma uncertainty range for 68% of the cases, i.e. 32% on average are expected outside the range. I suggest also assessing the statistical posterior uncertainty, and including these as error bars in the respective figures.

The appendix appears as another version of the explanation of the simulations, rather than only providing information that is additional to the main manuscript content. For example, eq. A1 and A2 of the appendix are identical to eq. 1 and eq. A2 of the main text.

Contradicting description of the MCMC method: In line 209 it is mentioned that the MCM method is applied without prior constraint (no regularization), then in line 225 it is

mentioned that assumed distributions of lambda_prior are used, which indicates that a prior constraint is used.

Detailed comments:

L45: "different atmospheric transports" -> "different atmospheric transport models"

L75: "CarbonTracker fossil fuel CO2" here a reference should be given

L83: add "Also" at the beginning of the sentence starting with "Other . . ."

L94-95: This is not a typical use of estimation error and uncertainty. If these terms are to be used to refer to synthetic and real data inversion, this should at least be made very clear. However I would not recommend using the terms that way.

L181: Note that CFM approaches can also involve simulations, at least when the number of unknowns is large (e.g. pixel-based inversions).

L189: add "The" before "Inversion"; also check throughout the manuscript for missing articles.

L233: the variance should have units corresponding to the square of the synthetic observations, i.e. if the observations are in ppm (for dry air mole fractions), the variance should have units ppm^2.

L289: Note that Gerbig et al. (2003) used temporal and spatial correlation in the measurement uncertainty related to transport error, thus their "D_epsilon" was not diagonal.

Fig. 5 caption: The numbers next to the symbols and the two rows of numbers in brackets below the x-axis should be mentioned/explained in the figure caption. Also the error bars shown in Fig. 5 (b) should be explained. Why are there no error bars in Fig. 5 (a)? What exactly is shown as the y-axis, is it the difference between posterior and target (truth) after spatial aggregation to the respective region (AB+SK and ON) and after temporally aggregated from monthly to annual? Why then are the error bars based on the standard deviation of the monthly errors, and not on the annual errors?

[Figure]

L384: "large degrees of freedoms" -> "large number of degrees of freedoms"

L396: "Bielgers" -> "Bieglers"

L444: "transport errors are in our experiments are" remove the first "are"

Figure 6: it should be mentioned which transport model is used in the pseudo observations. Either in the caption or in the text near line 460. I assume that CT2011 transport was used, corresponding to the case with prior flux and transport error.

L501, also L610-614: 32% of the estimates are expected to not include the truth within the 1-sigma uncertainty range, thus it is not required that all estimates include the truth within their uncertainty range.

L568: The sensitivity experiments should be added to table 3 so that it is clear which station was omitted in which experiment.

L669: The fact that aggregation error does not play an important role is due to the fact that target fluxes and prior fluxes are very close to each other in terms of spatial pattern. It should be clearly discussed as to how far this difference is expected to really represent differences between prior flux and true flux.

L693-696: this general statement on the nature of regional flux inversion should be backed up by references. Note that this statement is quite in contradiction to typical regional inversion results (see e.g. Lauvaux et al.2016).

L698-707: This discussion should include a discussion of pixel-based inversions (solving for spatially resolved fluxes at high resolution but using spatial (or temporal) correlation in the prior uncertainty) as it is state of the art nowadays.

Technical note: it would be much easier for reviewers if the captions for figures and tables were not separated from the figures and tables.

References:

Lauvaux, T., Miles, N. L. and Deng, A.: High‐resolution atmospheric inversion of urban CO2 emissions during the dormant season of the Indianapolis Flux Experiment (INFLUX), J. Geophys. Res., 121, doi:10.1002/2015JD024473, 2016.

———————————————

---

## Author Comment (AC1) · 10 Feb 2017

**Response to Anonymous Referee #1**

We thank Referee #1 for a detailed review of our paper, which certainly helped in improving the paper. In the following, we respond to the individual points raised in italic letters (in blue), the reviewer comments are left in normal font (in black).

This study investigates the performance of a regional inversion of anthropogenic  $CO_2$  emissions using synthetic experiments varying the prior fluxes, transport model and optimization method, and inversion setup. Conclusions are drawn regarding the optimal number of regions to be optimized, and the relative importance of transport model uncertainties. However, as will be explained further below, it is not clear what we learn in the end.

Authors' response: This is a model description and evaluation paper. What we learn from this study is that it is very possible to produce flux results with large estimation errors and unrealistic spatial patterns (negative fluxes where they should be positive) when the differences between the prior model concentrations and observations are large and/or biased.

The message that we would like to convey was that an extensive evaluation was necessary for any inverse modelling study (often not done) before the application of the model to real observations. Systematically increasing the number of unknowns (e.g. sub-regions as in our study) in combination with variations in the major model components (e.g. prior fluxes, model transport) is one way to check the stability of the inversion setup and the robustness of the results, as well as identifying the suitable setup for real observation inversion. Understanding the sensitivity of the flux estimates to prior error assumptions, region definitions, flux error, transport error, and optimization procedures could identify where to focus our effort on improving the component that was the most important in the inversion system. We will highlight what we have learned in the Abstract and Conclusions sections more clearly.

Most of the findings that are presented will depend on the specific setup of the inversion that is chosen. However, since this setup is far from realistic – the practical significance of the results for regional inverse modeling of  $CO_2$  using real data remains unclear. In my opinion, this point will have to be addressed clearly in the revised version of the manuscript to make this manuscript acceptable for publication.

Authors' response: it was not the intention for the authors to claim that the current version of the inverse model was ready to estimate regional  $CO_2$  emissions from fossil and/or biospheric sources. We will make it clear in the current version of our inverse model can be used to estimate slow varying monthly emissions as long as the transport error is small. In fact, we used fossil fuel  $CO_2$  and anthropogenic  $CH_4$ as two different priors in this study to show our inversion approach could work for both of these slow monthly varying and positive only emissions.

This comment regarding our model setup and the practical significance of the results is expanded to many detailed comments below. We will address these detailed comments in sequence and make clear that the inversion setup examined here is not intended for use to infer the total regional  $CO_2$  fluxes, please see below.

In addition, to improve readability I recommend that the authors focus only on the main findings, which could substantially reduce the size of the paper.

Authors' response: The descriptions of MCMC and CFM will be moved to the supplementary section. The Introduction, Section 2.2 the description of the prior fluxes and wherever possible will be shortened. In fact, a sensitivity analysis of the estimates to the prior flux (10%, 100%) and transport (10%, 30%, 100%) error assumptions was intentionally left out to reduce the size of the paper. As we found in this study, the specifications of the errors were not the determining factor.

**GENERAL COMMENTS**

The authors recommend that an inversion system is first tested in a synthetic environment before it is applied to real data. I fully agree to this, however, since the outcome of such an experiment depends on the specific details of the setup it should be realistic in the sense that the same setup could directly be applied to real data. This is clearly not the case for the setup that is presented here, since it only addresses anthropogenic emissions of  $CO_2$ , ignoring the natural component of the regional carbon cycle.

Authors' response: The model evaluation (and first real data inversion) was to start with the simple case of source only (positive fluxes, no sink or negative flux) to facilitate the evaluation. A suitable chemical species is wintertime  $CH_4$  as noted, when the wetland sources, soil sinks and OH chemical reaction (over the 5-day model integration period) are small or negligible in a regional sense.

We considered using CarbonTracker-  $CH_4$  (Bruhwiler et al., 2014) as the target result. However, the low resolution of  $CT-CH_4$  transport model (TM5 with 4 deg latitude x 6 deg longitude) would not be sufficient to fully resolve the four measurement sites or more than a few sub-regions in our inversion domain (~ 10 deg latitude x 20 deg lon longitude).

A more suitable choice was CarbonTracker fossil fuel  $CO_2$  (CarbonTracker, 2010, 2011), with positive fluxes and varying on monthly timescale (or without the added complexity of diurnal variations, the typical assumption for  $CH_4$  inversion studies). CarbonTracker  $CO_2$  has the higher resolution of 1 deg x 1 deg for North America (including our domain and surrounding).

The prior flux spatial distribution of CT-ff is similar (not identical) to the anthropogenic  $CH_4$  (our target first real observation inversion, in progress) with some point sources (large facilities). Therefore we used CT-ff as the target for flux and mixing ratios which allow for the testing of prior flux errors, transport errors, background mixing ratio errors, and so on.

These explanations will be included in the revision of the manuscript.

In addition, the boundary conditions of the regional domain that is optimized are assumed to be known exactly. Since the regional biospheric fluxes are expected to be the main uncertain component, it is unclear to me how this inversion is supposed to work when applied to real data.

Authors' response: Although we did not optimize the boundary conditions due to the fact that there were not enough observations and the estimation errors were already large, we made a note in the text that there could be errors associated with the boundary conditions (estimated baseline values). Therefore we made no distinction between boundary error and regional transport error. A general term of "transport error" was used to encompass all errors associated with transport modelling.

It is mentioned that the methodology could be applicable to wintertime  $CH_4$  fluxes. But if this is the application that the authors have in mind then why perform a test inversion for fossil  $CO_2$  instead of  $CH_4$ ?

Authors' response: At the time of conducting this study and model development, the finest possible horizontal resolution of predicted greenhouse gas concentrations was provided by CarbonTracker  $CO_2$  at  $1^{\circ}x1^{\circ}$ , whereas the output resolution of CarbonTracker  $CH_4$  was  $6^{\circ}x4^{\circ}$ . It was not possible to use  $CT-CH_4$  as the target.

The authors comment on the estimated posterior uncertainties in relation with the actual deviations from the predefined true fluxes, concluding that the estimates are too optimistic. However, this conclusion depends on how the assumed a priori flux and data uncertainties reflect the actual errors. It seems that no effort was made to analyze the statistics of the difference between for example CT2011 and CT2010 before specifying the a priori flux covariance. The same is true for the model-data mismatch.

Authors' response: We did not go into the details to study the statistics of the difference between CT2011 and CT2010 before specifying the prior flux error covariance. Only monthly simple statistics were calculated as presented in Table 2.

Instead, we tested different a priori flux and model-data mismatch uncertainties using 10% and 100% for both  $(\sigma_{prior})^2$  and  $(\sigma_e)^2$  and found that the flux estimates were not sensitive to a priori and model-data uncertainties on the provincial and annual time scales. This result could be included in the revision, but it was intentionally left out to shorten the manuscript.

When the transport model error is small, it was not necessary to concern with the a priori uncertainties as shown in Figure 5 (flux error case). When the transport model error is large, specifying different prior uncertainties unfortunately cannot remove such systematic model error.

In practice, we often do not know the true prior flux and model-data mismatch errors and the spatiotemporal structures, and most important, the interaction of the flux and model-data mismatch errors as all sources of errors are folded together in the modelled mixing ratios. Therefore, we focus our effort on carrying out a detailed assessment of the sensitivity of the estimates to different assumptions and setups for inversions. Figures C1 and C2 below show that 100% prior flux error used throughout should be liberal enough to allow the parameter estimates to be calculated on a wide possible range for the seven region definitions used in our study.

Figure C1 shows the relative percentage difference between CT2011 and CT2010 for the AB+SK inversion domain. The prior flux errors range from +3% to -79% as shown on the map depending on the region definitions.

---

## Author Comment (AC2) · 10 Feb 2017

**Response to Anonymous Referee #2**

We thank Referee #2 for the constructive assessment of our paper. In the following, we respond to the individual points raised in italic letters (in blue), the reviewer comments are left in normal font (in black).

The manuscript describes a regional inverse modeling system that uses atmospheric observations to estimate GHG fluxes. Synthetic experiments are performed to evaluate the system. A number of specific aspects need to be addressed before I can recommend accepting the manuscript for publication.

**Main comments:**

Regarding the number of regions: I fully agree with referee #1 in that a prior error of 100% for different regions with a changing number of regions will result in a decrease of the uncertainty of the spatially integrated fluxes, as the errors are assumed uncorrelated.

One possible method would be to inflate the variance accordingly such that the prior error of the spatially aggregated flux does not change with the number of regions. However this has an impact on the comparisons between flux estimates using a different number of regions to be solved for.

Authors' response: In our results, increasing the number of sub-regions can increase or decrease the total (sum of the sub-regions) posterior error, and similarly increase or decrease the posterior uncertainties. The optimization procedures can behave in a complex way with the number of sub-regions.

For example, it is possible for the atmospheric transport to correlate the flux signals from different subregions. If a parcel trajectory passed through different sub-regions with non-zero fluxes over the 5-day period, then the flux signals from the different sub-regions could be mixed (correlated) within the parcel. Integrating the contributions from all the parcels, the flux errors for the different sub-regions in combination with transport errors could be correlated in complex ways (particularly when the errors are biased).

As it is likely not the case in our model that the 'errors are uncorrelated', the posterior uncertainty of the whole region in our results could go up or down as it is divided up into a larger number of sub-regions (and not always 'go down').

These characteristics in the posterior uncertainties can be seen more clearly in the revised version (see below) of Figure 5 in which the scales of the y-axes have been changed to improve readability. The number of negative sub-regions for the corresponding number of sub-regions is shown on the 2nd row from the bottom in the figure. These explanations will be added to the manuscript revision.

This points to the value of evaluating the inversion model for many different setups to assess the stability and robustness of the inversion model.

**MCMC**

**CFM** Alberta+Saskatchewan (AB+SK) (I) Prior flux error (II) Transport error (III) Prior flux and transport error (Section 3.1) (Section 3.2) (Section 3.3) 30 Annual Error (%) 20 16 5 15 10 0 -10 -20 10 11 12 13 14 15 16 17 18 19 20 21 9 2 3 5 6 8 1 4 7 Experiment # of negative sub-regions in 12 months (0) (0) (0) (0) (0) (0) (0) (0) (6) (9) (9) (24) (45) (48) (0) (3) (3) (3) (18) (33) (36) # of sub-regions (2) (4) (7) (11) (19) (27) (37) (2) (4) (7) (11) (19) (27) (37) (2) (4) (7) (11) (19) (27) (37) b